# Use of varenicline and nicotine replacement therapy in people with and without general practitioner-recorded dementia: retrospective cohort study of routine electronic medical records

Taha Itani,[1,2] Richard Martin,[1,3,4] Dheeraj Rai,[3,5,6] Tim Jones,[7] Gemma Taylor,[8] Kyla Thomas,[4] Marcus Munafo,[1,2] Neil Davies,[1,4] Amy Taylor[3,4]

For numbered affiliations see end of article.

**Correspondence to**
Dr Taha Itani;
ti17926@bristol.ac.uk

## ABSTRACT

**Objectives** Our primary objective was to estimate smoking prevalence and prescribing rates of varenicline and nicotine replacement therapy (NRT) in people with and without general practitioner (GP)-recorded dementia. Our secondary objective was to assess and compare quit rates of smokers with versus without GP-recorded dementia who were prescribed varenicline or NRT for smoking cessation.

**Design** A retrospective cohort study based on the analysis of electronic medical records within the Clinical Practice Research Datalink (2007–2015).

**Setting** 683 general practices in England.

**Participants** People with and without GP-recorded dementia, aged 18 years and have a code indicating that they are a current smoker.

**Intervention** Index prescription of varenicline or NRT (from 1 September 2006).

**Outcome measures** The primary outcomes were smoking prevalence and prescribing rates of varenicline and NRT (2007–2015). The secondary outcome was smoking cessation at 2 years.

**Results** Age and sex-standardised prevalence of smoking was slightly higher in people with GP-recorded dementia than in those without. There were 235 314 people aged 18 years and above prescribed NRT or varenicline. Among smokers with GP-recorded dementia (N=447), 409 were prescribed NRT and 38 varenicline. Smokers with GP-recorded dementia were 74% less likely (95% CI 64% to 82%) to be prescribed varenicline than NRT, compared with smokers without GP-recorded dementia. Compared with people without GP-recorded dementia, people with GP-recorded dementia had consistently lower prescribing rates of varenicline from 2007 to 2015. Two years after prescription, there was no clear evidence for a difference in the likelihood of smoking cessation after prescription of these medications between individuals with and without dementia (OR 1.0, 95% CI 0.8 to 1.2).

**Conclusions** Between 2007 and 2015, people with GP-recorded dementia were less likely to be prescribed varenicline than those without dementia. Quit rates following prescription of either NRT or varenicline were similar in those with and without dementia.

## Strengths and limitations of this study

► This study used primary care data from the Clinical Practice Research Datalink (CPRD) which are representative of the UK primary care population.
► Expert-reviewed codelists were developed to define both the exposure and the outcome which would reduce misclassification bias.
► Due to the small sample size of people with general practitioner-recorded dementia, we were not able to test the relative effectiveness of varenicline versus nicotine replacement therapy (NRT) on smoking cessation using regression models.
► Data on smokers who purchase over-the-counter NRT were not available, and therefore the prevalence of NRT might be underestimated.

## INTRODUCTION

Smoking is a leading cause of mortality and morbidity worldwide. About 12% of global deaths were linked to smoking in 2015.[1] There is substantial evidence that smoking is associated with an increased risk of developing dementia.[2 3] For instance, it is estimated that 14% of Alzheimer's disease (AD) cases worldwide are attributable to smoking.[4] Smoking is thought to accelerate the onset of dementia mainly via vascular risk factors such as narrowing of blood vessels in the heart and the brain, thereby triggering oxidative stress.[4 5]

Few studies report smoking prevalence among people with dementia. In a cross-sectional analysis of patients treated for AD in a neurology clinic during a 2-year period, previous smoking prevalence was 29% (N=21/72).[6] In a case–control study of patients with vascular dementia, the rate of current tobacco use was 9% (N=17/198) as compared with 6% (N=11/199) in the control

group.[7] Beyond the harmful health effects of smoking in people with dementia, there are concerns that smokers in this group may have a higher likelihood of fire accidents due to their compromised cognitive state.[8]

Since there are currently no available treatments to cure dementia, there is a growing interest in identifying modifiable risk factors for reducing the occurrence of the disease, to delay dementia onset, and reduce its burden.[4 9] Smoking cessation could potentially decrease the risk or slow the development of dementia[10] and could improve the quality of life of older adults through improved physical, and mental well-being.[9 11] Little is known about whether people with dementia are prescribed smoking cessation agents and whether they are effective in this group.

Based on a Cochrane review of 136 trials, it was reported that nicotine replacement therapy (NRT) (compared with placebo or no treatment) can help people who make a quit attempt to increase their chances of successfully stopping smoking (Hartmann-Boyce *et al*).[12] Data from observational studies[13 14] and meta-analyses of randomised controlled trials[15] indicate that varenicline is more effective than single form NRT for smoking cessation in the general population. However, it is unclear whether varenicline or NRT could help smokers with dementia to quit smoking.

Therefore, in this study we aimed to (1) describe the rates of smoking prevalence and smoking cessation medication prescribing among people with and without general practitioner (GP)-recorded dementia in UK primary care settings from 2007 to 2015 and (2) assess and compare associations of varenicline and NRT on smoking cessation in people with GP-recorded dementia, compared with those without, at 3, 6, 9 months and 1, 2, 4 years after first prescription.

## METHODS
### Data source and population
We conducted a retrospective cohort study using electronic medical records from 683 general practices in England from 2007 to 2015 using data from the Clinical Practice Research Datalink (CPRD). Patient data from the CPRD are broadly representative of the UK primary care population in terms of age, sex and ethnicity.[16] These data have been validated, audited and quality checked.[17] The study's protocol (15_115R) was approved by the Independent Scientific Advisory Committee for MHRA Database Research (https://www.cprd.com/isac/).

### Code lists
We defined variables using medical and product codes within the CPRD. All code lists were developed using a list from a previously published study.[13] These codelists were derived from the the British National Formulary (BNF) and the International Classification of Diseases (ICD-10) and then agreed on by field experts (DR, KHT). A previous systematic review that checked the validity of coding of

various diagnoses in what was then the General Practice Research Database (now CPRD) suggests that coding for dementia and Alzheimer's is relatively accurate.[18]

### Study subjects
During the study period (2007–2015), we included people (aged ≥18 years) with information about their smoking status (either smokers or non-smokers) for smoking prevalence estimates, and we included smokers prescribed either varenicline or NRT for prescribing prevalence and for comparing quit rates. We used an open cohort design, with new patients entering the cohort throughout the observation period.

For the primary objective, people were categorised as having dementia if: (i) they had ever been diagnosed with dementia (based on ICD-10 diagnoses F00-F03), or (ii) if they were prescribed dementia medications: (BNF chapter 4.11) (see online supplementary eTables 1 and 2 for a list of all Read and product codes used in this study). Then, the earliest record of GP-recorded dementia in the CPRD was taken forward ensuring that all records used were within the registration period of each patient. Patients with no records of the above-mentioned diagnosis/prescriptions were considered to have no GP-recorded dementia (for clarity, we hence forth refer to this as dementia).

For the secondary objective, we constructed a cohort of eligible first varenicline/NRT prescriptions (see online supplementary eFigure 1 for a flowchart of numbers of patients excluded and reasons for exclusion). Within that cohort of eligible prescriptions, we considered individuals with dementia to be those with recorded Read codes for ever dementia/dementia medications prior to first varenicline/NRT prescription; we did this to ensure that a diagnosis of dementia preceded the exposure (prescription of a smoking cessation medicine).

### Variables
#### Smoking and prescribing prevalence estimates
For smoking prevalence estimates, a patient's smoking status (aged ≥18 years) was defined by a record indicating smoking/non-smoking or prescription of NRT/varenicline in that year. In case of missing information about smoking, the patient's smoking status was carried forward until there was evidence of a change in smoking status or carried backwards if smoking status was only recorded in the final year of registration. Records that were outside the registration period for each patient were excluded.

### Exposure
Exposure was defined as prescription of varenicline or NRT (eg, patches and so on, on prescription as opposed to over-the-counter, hence forth we refer to this as NRT).

Prescriptions used to define exposure groups were issued between 1 September 2006 and 31 August 2016, with no prior record of use of a related product in the preceding 18 months. We used the first treatment episode to ensure that intervention groups were 'new users' of the

medication.[19] We did not model multiple and repeated prescriptions of smoking cessation medications during follow-up because this is likely to be strongly related to patient characteristics.

## Outcome: smoking cessation

Smoking cessation was defined as having an electronic record indicating a non-smoking status. The closest smoking record to each follow-up period was selected to determine each study participant's smoking status; that is, the most recent smoking record identified between cohort entry and each follow-up period (eg, 3 months, 6 months). People with missing smoking data (beyond 180 days) were assumed to be continuing smokers[20] which has been previously found to be robust in sensitivity analyses.[13]

## Covariates

Covariates included patients' age at time of prescription, sex, index of multiple deprivation score (IMD), mean number of GP visits 1 year prior to first prescription, year of first prescription of a smoking cessation medication, body mass index (BMI), days registered in the CPRD, the Charlson Index (a measure of chronic illness),[21] alcohol misuse, history of mental disorder or psychoactive medication prescriptions, evidence of other psychoactive medication prescription or other less common psychiatric disorders. We used multiple imputation to handle missing data on BMI and IMD. This was done using the ICE command in Stata where we produced 20 imputed datasets (see online supplementary eTables 3). We included all exposures, covariates and outcomes in the imputation model.[22]

## Follow-up

The secondary outcome was smoking cessation at 2 years, and this was also assessed at 3, 6, 9 months and 1 and 4 years after first prescription of varenicline or NRT.

## Statistical analysis
### Smoking prevalence

Smoking rates were calculated by dividing the number of people with dementia who had Read codes indicating current smoking for each year between 2007 and 2015 by the total number of people with dementia and a smoking status code (indicating current or non/ex-smoking) each year between 2007 and 2015. For comparison, smoking prevalence was also estimated among people without dementia.

### Prescribing prevalence

The prevalence of varenicline and NRT prescribing among current smokers was calculated by dividing the number of prescriptions each year from 2007 to 2015 (there were no varenicline prescriptions for patients with dementia in 2006) by the number of current smokers in each year. Prevalence was estimated for people with and without dementia. Individuals with missing smoking information were excluded from the denominators.

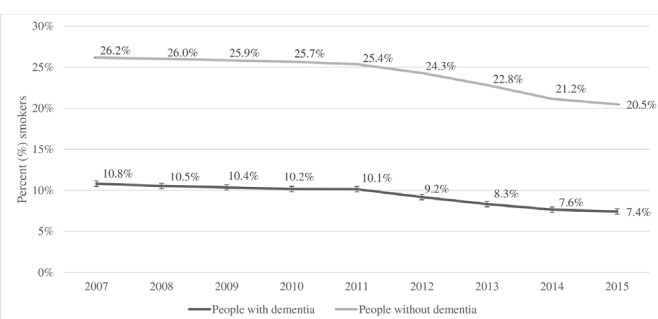

**Figure 1** Percentage (%) of primary care patients with an electronic medical record indicating smoking, from 2007 to 2015, in people with or without dementia.

## Association of varenicline and NRT prescriptions with smoking cessation

We had originally planned to compare the effectiveness of NRT and varenicline for smoking cessation in individuals with dementia. However, given the small numbers of individuals prescribed varenicline, we had insufficient power to conduct this analysis. Therefore, we compared the effectiveness of being prescribed either varenicline or NRT on smoking cessation in individuals with dementia compared with individuals without dementia. This was determined by estimating quit rates at each follow-up period for individuals prescribed either of these medications. This was calculated by dividing the number of non-smokers in each group by the total number of people in that group at each follow-up period. All analyses were conducted using Stata V.14 MP.

## Patient and public involvement

This study was based on the analysis of anonymised primary care data. No patients were involved during the design and analysis of this study.

## RESULTS
### Smoking prevalence and smoking cessation medication prescribing estimates

Unadjusted smoking prevalence among people with dementia steadily decreased from 11% (N=2965/27 432 in 2007 to 7% (N=2690/36 249) in 2015 (figure 1). These estimates were consistently lower than in people without dementia, 26% (N=1 010 530/3 860 169) in 2007 and 21% (N=628 444/3 068 743) in 2015, respectively (see online supplementary eTable 4). However, after age and sex standardisation, the smoking prevalence among people with dementia was slightly higher than in people without (see online supplementary eFigure 2).

The rate of NRT prescribing in people without dementia was 7% (68 935/1 010 530) in 2007 which decreased to 2% (N=13 626/628 444) by 2015, whereas NRT prescribing rates among people with dementia increased during the same period. Compared with people without dementia, people with dementia had lower prescribing rates of varenicline (figure 2).

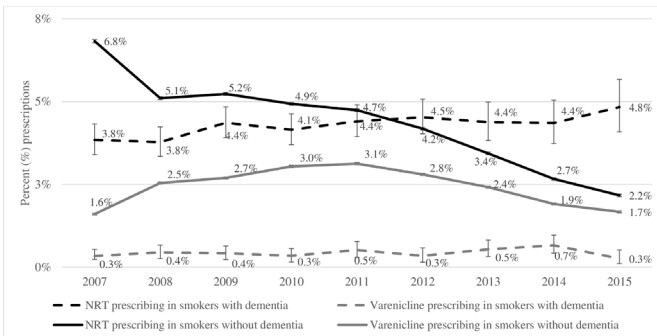

**Figure 2** Prescription prevalence of varenicline or NRT in primary care, from 2007 to 2015, in smokers with dementia, compared with smokers without dementia. NRT, nicotine replacement therapy.

## Smoking cessation amongst individuals prescribed NRT and varenicline

Of the 235 314 people included in this analysis, 447 were people with dementia, whereas 234 867 were people without (see online supplementary eFigure 1). Overall, 159 736 smokers were prescribed NRT and 75 578 prescribed varenicline (table 1). The mean age of people with dementia at the time of smoking medication prescription was about 72 years (SD=12.6), while that of people without dementia was 46 years (SD=14.8) People with dementia were about 25 years older, had more GP visits 1 year prior to the first prescription, suffered from more comorbidities and received more mental health-related prescriptions than those without dementia (table 1).

Smokers with dementia were 74% (95% CI 64% to 82%) less likely to be prescribed varenicline than NRT, compared with smokers without dementia (see online supplementary eTable 5). The proportion of people with and without dementia who quit smoking after being prescribed either varenicline or NRT increased throughout the study's follow-up period (3 months to 4 years). At 2-year follow-up, people with dementia were more likely to quit smoking (30.6%, 95% CI 25.8% to 35.1%) than those without (25.7%, 95% CI 25.4% to 25.8%) when prescribed either varenicline or NRT (figure 3) (see online supplementary eTable 6). However, after adjusting for all covariates, we found no evidence for a difference in quit rates between individuals with and without dementia (OR=1.0, 95% CI 0.81 to 1.23) (see online supplementary eTable 7).

## DISCUSSION

People with dementia were less likely to be prescribed varenicline compared with those without dementia. There was no clear evidence for a difference in quit rates in individuals with and without dementia following prescription of NRT or varenicline.

A strength of this study is that we used primary care data from the CPRD which are representative of the UK primary care population.[16] Hence, smoking rates in people with dementia in this study are likely to be generalisable to the

dementia population in the UK and in similar countries. Additionally, we used expert-reviewed codelists to define both the exposure and the outcome which would reduce misclassification bias (ie, classifying people with dementia as people without and vice versa).

There are several limitations to this research. Due to the small sample size of people with dementia, we were not able to test the relative effectiveness of varenicline versus NRT on smoking cessation using regression models. We had no data on smokers who purchase over-the-counter NRT, therefore we might be underestimating the prevalence of NRT use, particularly among people without dementia. Hence, it is likely that the prevalence of NRT use among people without dementia is larger than the prevalence of NRT prescribing in this study. Moreover, outcome definition (smoking vs non-smoking status) was based on self-reported data rather than biochemical verification of smoking status. Additionally, social desirability bias may occur when unsuccessful quitters don't disclose their smoking status truthfully. We also relied on point estimates (ie, smoking status reported at a single time-point) for making conclusions about smoking status. This may not have captured long-term abstinence. In other words, it is possible that smoking status may have fluctuated between GP visits. A further limitation is having no information about patient adherence in taking their prescribed smoking cessation medications.

We are not aware of previous population-based studies that estimated the smoking rates among people with dementia as the available evidence has been limited to small cross-sectional studies. For instance, a community study in China found that about 17% (N=69/186) of the elderly sample with dementia were current smokers compared with 25% (N=415/1664) in people without dementia.[23] Results from the Toyama dementia survey in Japan show that only 4% of people with dementia smoked compared with 10% in those without.[24] This high variability in the results points to the need for larger and more representative studies in people with dementia to be conducted. Meanwhile, we found that smoking prevalence has decreased steadily among people without dementia, from 26% in 2007 to 21% in 2015. This was fairly similar to the general population in England as evidenced by results from the Smoking Toolkit Study (24.2% in 2007 to 18.7% in 2015)[25] and therefore speaks to the external validity of our study.

We observed a low prevalence of varenicline prescribing during our study period in people with dementia. Our estimate for individuals without dementia was consistent with findings from a previous study that examined the use of varenicline for smoking cessation treatment in UK primary care using data from THIN database in 2011. Compared with our results from that year, our estimates appear slightly lower (1.1% vs 1.8% in the THIN study).[26] While NRT prescribing rates increased from 4% in 2007 to 5% in 2015 in people with dementia, these rates declined over time in people without dementia. A recent report from the British Lung Foundation found

**Table 1** Baseline characteristics of people with or without dementia by exposure group, N (%)

| Characteristic | People with dementia (N=447) | | | People without dementia (N=234867) | | |
|---|---|---|---|---|---|---|
| | NRT (N=409) | Varenicline (N=38) | Total | NRT (N=159327) | Varenicline (N=75540) | Total |
| Age at time of first prescription* | 71.1 (12.2) | 66.2 (15.1) | 70.7 (12.6) | 46.2 (15.5) | 44.4 (13.2) | 45.6 (14.8) |
| Sex (male) | 186 (45.5%) | 19 (50.0%) | 205 (45.9%) | 73 674 (46.2%) | 37 676 (49.9%) | 111 350 (47.4%) |
| Index of multiple deprivation score (IMD)†‡ | 3 | 4 | 3 | 3 | 3 | 3 |
| No of GP visits 1 year prior to first prescription* | 11.5 (9.0) | 15.3 (9.9) | 11.8 (9.1) | 8.9 (7.4) | 7.3 (6.1) | 8.4 (7.0) |
| BMI*† | 24.6 (5.1) | 25.7 (6.3) | 24.7 (5.4) | 26.5 (5.7) | 26.5 (5.4) | 26.5 (5.6) |
| Year of first prescription† | 2010 | 2010 | 2010 | 2009 | 2010 | 2009 |
| Days of history* | 3573.8 (2181.2) | 3450.3 (2327.4) | 3563.3 (2191.5) | 3052.9 (1907.1) | 3164.8 (1986.2) | 3088.9 (1933.6) |
| Comorbidity ever (Charlson Index) | 354 (86.6%) | 28 (73.7%) | 382 (85.5%) | 59 489 (37.3%) | 24 017 (31.8%) | 83 506 (35.6%) |
| Alcohol misuse ever | 104 (25.4%) | 11 (29.0%) | 115 (25.7%) | 13 890 (8.7%) | 4759 (6.3%) | 18 649 (7.9%) |
| Self-harm ever | 67 (16.4%) | 9 (23.7%) | 76 (17.0%) | 17 232 (10.8) | 6652 (8.8%) | 23 884 (10.2%) |
| Ever anxiety and stress related disorders | 151 (36.9%) | 16 (42.1%) | 167 (37.4%) | 44 381 (27.9%) | 17 377 (23.0%) | 61 758 (26.3%) |
| Other behavioural/neurological disorder ever | 30 (7.3%) | 6 (15.8%) | 36 (8.1%) | 8693 (5.5%) | 2956 (3.9%) | 11 649 (5.0%) |
| Ever depression | 217 (53.1%) | 26 (68.4%) | 243 (54.4%) | 65 343 (41.0%) | 26 097 (34.6%) | 91 440 (38.9%) |
| Ever antidepressants | 273 (66.7%) | 28 (73.7%) | 301 (67.3%) | 79.584 (50.0%) | 32 230 (42.7%) | 111 814 (47.6%) |
| Ever antipsychotics | 175 (42.8%) | 13 (34.2%) | 188 (42.1%) | 28 972 (18.2%) | 9792 (13.0%) | 38 764 (16.5%) |
| Ever hypnotics/anxiolytics | 238 (58.2%) | 20 (52.6%) | 258 (57.7%) | 60 092 (37.7%) | 25 134 (33.3%) | 85 226 (36.3%) |

*Data presented are mean and SD.
†Data presented are median.
‡Missing data: BMI data were missing for 14.1% (N=33 059); IMD data were missing for 43.6% (N=102 657).
BMI, body mass index; GP, general practitioner.

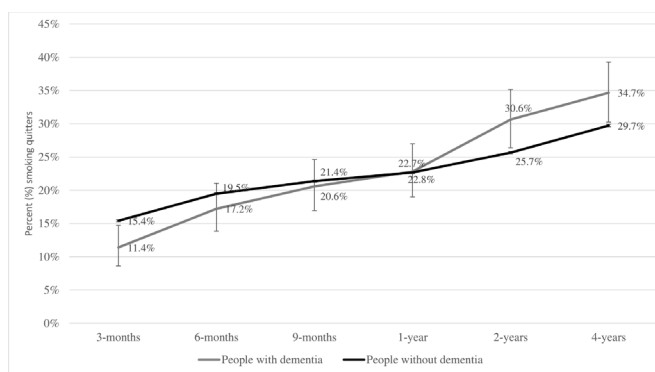

**Figure 3** Percentage (%) of people with an electronic medical record indicating smoking cessation at up to 2 years follow-up, in people with and without dementia.

that NRT prescribing through primary care in England has dropped about 75% during the last 10 years. That was mainly due to cuts to public health funding that would have adversely impacted specialist stop smoking services.[27]

Our study is among the first to investigate longer term smoking cessation after being prescribed varenicline and NRT among individuals with dementia in a real world setting. Our results suggest that both varenicline and NRT could produce long-term smoking cessation in people with dementia as they do in those without. Almost one-third of smokers with dementia quit smoking after 2-year follow-up. Regardless of the smoking cessation medication prescribed to people with dementia, it is important to acknowledge that achieving smoking cessation in this group may carry health benefits which would potentially improve their general health status and and/or extend life expectancy.[28] This should ideally be coupled with improving diet quality and increasing physical activites that may shield quitters from weight gain after smoking cessation.[29]

It is not clear why individuals with dementia are less likely to be prescribed varenicline than NRT compared with individuals without. Previous clinical and observational studies have established that varenicline is superior to single form NRT in achieving smoking cessation in different groups.[13 30 31] Additionally, varenicline did not seem to be associated with an increased risk of documented cardiovascular events, depression or self-harm when compared with NRT in primary care in England.[32] On the other hand, a recent study based on CPRD data concluded that NRT appears to increase cardiovascular events for patients prescribed NRT, compared with those receiving smoking cessation advice after 52 weeks of follow-up.[33] This was consistent with the evidence shown by a meta-analysis of 120 studies involving 177 390 individuals.[12] It is possible that GPs are less likely to prescribe varenicline to individuals with dementia because of lower likelihood of adherence; in a recent systematic review of the literature, older patients with dementia were found to have a low level of medication adherence.[34]

In summary, age-adjusted and sex-adjusted smoking prevalence among individuals with dementia was similar to those without dementia and smoking cessation rates were similar following prescription of smoking cessation medications between these groups. However, individuals with dementia were less likely to be prescribed varenicline than individuals without dementia.

**Author affiliations**
[1]Medical Research Council Integrative Epidemiology Unit, University of Bristol, Bristol, UK
[2]UK Centre for Tobacco and Alcohol Studies, School of Psychological Science, University of Bristol, Bristol, UK
[3]NIHR Biomedical Research Centre at the University Hospitals Bristol NHS Foundation Trust and the University of Bristol, Bristol, UK
[4]Population Health Sciences, Bristol Medical School, University of Bristol, Bristol, UK
[5]Centre for Academic Mental Health, School of Social and Community Medicine, University of Bristol, Bristol, UK
[6]Avon & Wiltshire Partnership NHS Mental Health Trust, Bristol, UK
[7]National Institute for Health Research Collaboration for Leadership in Applied Health Research and Care West (NIHR CLAHRC West) at University Hospitals Bristol NHS Foundation Trust, Bristol, UK
[8]Addiction and Mental Health Group (AIM) Department of Psychology, University of Bath, Bath, UK

**Contributors** TI contributed to data cleaning, data analysis, interpretation of results and writing the manuscript. RM, GT, ND, AT and KT contributed to study conceptualisation, study design, interpretation of results, data analysis and writing the manuscript. MM and DR contributed to study conceptualisation, study design, interpretation of results and writing the manuscript. TJ extracted the data and contributed to writing the manuscript. TI, AT and ND had full access to all of the data in the study and takes responsibility for the integrity of the data and the accuracy of the data analysis.

**Funding** KT is funded by a National Institute for Health Research postdoctoral fellowship (PDF-2017-10-068). TJ receives funding from the National Institute for Health Research Collaboration for Leadership in Applied Health Research and Care (NIHR CLAHRC) West. The views expressed are those of the authors and not necessarily those of the NHS, the NIHR or the Department of Health. RM is supported by the NIHR Bristol Biomedical Research Centre, a partnership between the University Hospitals Bristol NHS Foundation Trust and the University of Bristol; and by a Cancer Research UK Programme Grant (C18281/A19169). GT is funded by a Cancer Research UK Population Researcher Postdoctoral Fellowship Award (C56067/A21330). This research was supported by Global Research Awards for Nicotine Dependence (GRAND), an independently reviewed competitive grants program supported by Pfizer, to the University of Bristol.

**Disclaimer** The funder had no role in study design, data collection and analysis, decision to publish or preparation of the manuscript.

**Competing interests** None declared.

**Patient consent for publication** Not required.

**Provenance and peer review** Not commissioned; externally peer reviewed.

**Data availability statement** Data may be obtained from a third party and are not publicly available.

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
