## [Reviewer comments · BMJ Open]

ARTICLE DETAILS

TITLE (PROVISIONAL)	Use of varenicline and nicotine replacement therapy in people with and without general practitioner-recorded dementia: retrospective cohort study of routine electronic medical records.
AUTHORS	Itani, Taha; Martin, Richard; Rai, Dheeraj; Jones, Tim; Taylor, Gemma; Thomas, Kyla; Munafò, Marcus; Davies, Neil; Taylor, Amy

VERSION 1 - REVIEW

REVIEWER	Emily Peckham University of York, UK
REVIEW RETURNED	22-Nov-2018

GENERAL COMMENTS	Thank you for asking me to review this article, which I enjoyed reading. I only have a few minor comments. 1. It might be useful to clarify why there was no need for an ethical review for this study2. The article would benefit from a greater discussion on the limitations of the study3. It would be useful to include for the mean age of people without dementia when prescribed smoking medication be included in the text.4. The fact that NRT rates declined in people without dementia but remained the same in people with dementia is interesting. Could this be picked up in the discussion?
---

REVIEWER	Daniel Kotz Institute of General Practice Addiction Research and Clinical Epidemiology Unit Medical Faculty of the Heinrich-Heine-University Düsseldorf Postal address: P.O. Box 101007, 40001 Düsseldorf, Germany
REVIEW RETURNED	29-Nov-2018

GENERAL COMMENTS	Itani et al. present findings from a GP database analysis regarding use of NRT and varenicline in smokers with GP-recorded dementia. The study addresses a very relevant topic and aims to fill a gap in the literature. Major strengths of the study include use of a large GP database, which has been used for similar research before, and prior publication of a detailed protocol. The manuscript
--

is well organised, but there are several issues which need to be addressed.

Major comments

1. The study is by no means a prospective cohort study but a retrospective cohort study. This should be changed throughout the manuscript. Also, please state whether you used an open cohort design; did new patients enter the cohort throughout the observation period? This is also not clear.

2. The aim of the study is unclear at various places in the manuscript (incl. the abstract). The aim was not to determine effectiveness, as varenicline was not compared to a control group (the natural course of quitting was not taken into account). It seems that this was the aim at the time of designing the study (in the study protocol it says: "What is the effect of varenicline on smoking abstinence..."), but it turned out not to be feasible to reach that aim. In the current manuscript, quit rates are reported (but not compared) in smokers who used varenicline or NRT, but one cannot draw valid conclusions regarding effectiveness of either varenicline or NRT by this approach. The aim of the current manuscript seems more to assess and compare quit rates of smokers with vs. without dementia who used varenicline or NRT for quitting.

3. Following the previous comment, the writing of the manuscript should be critically revised throughout with regard to the interpretation of findings. This is particularly important as the authors have a conflict of interest; the current research is funded by Pfizer, the manufacturer of varenicline. Examples:

a. The first sentence of the discussion is misleading as "after being prescribed" suggests some form of effectiveness assessment. The same regards the first sentence of the summary: "In summary, people with dementia are more likely to quit smoking when prescribed either varenicline or NRT, as compared to those without dementia".

b. The second sentence of the discussion is also misleading as lower rates were only found in year 2 and 4 of the observation period.

c. The following conclusion is invalid: "Our results suggest that both varenicline and NRT are effective in producing long term smoking cessation in people with and without dementia". Also, beware that "Almost one third of smokers with dementia quit smoking after 2-years follow-up" could be simply explained by natural course.

d. The second sentence of the summary should be revised: "... we found that people with dementia were less likely to be prescribed varenicline *and NRT* than people without dementia."

4. The rationale behind the study, in particular the focus on varenicline should be explained in more detail, preferably in the introduction (last but one paragraph). Elaborate on the effectiveness of other smoking cessation treatments, in particular NRT, and explain the rationale for focussing on varenicline. The problem statement "the effectiveness of varenicline for smoking cessation amongst people with dementia remains unknown" is a bit problematic as you are not able to assess this effectiveness (see comment above).

5. An obvious limitation of the current study – inherent to the study design – is the outcome definition. Lack of biochemical verification of non-smoking is problematic. For example, social desirability may be a source of bias when people with a new diagnosis of

dementia return to their GP without having successfully quit smoking. Also, it is not clear if point prevalence or prolonged abstinence rates are presented. If the former, please explicitly state so. Finally, there are no data on adherence to the smoking cessation medications – this is also a major limitation when interpreting the findings. All these issues should be discussed more thoroughly.

6. Regarding the prescription medication, please specify if and how you have accounted for multiple and repeated prescriptions of smoking cessation medications during follow-up.

7. Some results seem a bit odd to me; please check if these are correct.

a. (Page 12, lines 12-17) Why was the denominator in 2015 (n=42,075) so much lower than in 2007 (n=84,647)? In comparison, why were the denominators about the same in both years in the no-dementia group?

b. (Page 12, lines 24-25) Again, why was the denominator in 2015 (n=628,116) so much lower than in 2007 (n=1,007,563)?

8. I appreciate very much that a detailed protocol was published a priori. In the methods section of the current paper, I suggest to add a brief paragraph highlighting the most relevant deviations from the earlier protocol (see comments above).

9. Please provide a full list of all Read codes used in this study as supplementary material.

Minor comments

10. Correct (see earlier comments) and specify the title.
Suggestion: "Use of varenicline and nicotine replacement therapy in people with and without general practitioner-recorded dementia: retrospective cohort study of routine electronic medical records"

11. Specify "NRT on prescription (NRT Rx)" (as opposed to over-the-counter) throughout the manuscript.

12. Specify "GP-recorded dementia" throughout the manuscript.

13. Abstract:

a. Specify the index date of participants.

b. Fit secondary outcome measures with secondary objectives.

c. Prescribing rates should be adjusted for potential confounders, in particular age and co-morbidities; otherwise one cannot say that differences are due to dementia.

14. Strengths and limitations of this study: I know CPRD, it's a very good database, but certainly not "representative of the UK population." Rather, the data are "representative of the UK primary care population."

15. Introduction: "Little is known about whether people with dementia are prescribed smoking cessation agents ..." It would be helpful to mention available evidence-based treatments here.

16. Results, last sentence: rewrite the sentence to make clear that "versus" refers to the comparison of people with vs. without dementia.

17. Discussion: please compare the smoking prevalence rates in the no-dementia group with rates in the general UK population from other studies such as the Smoking Toolkit Study. This should add confidence to the external validity of the study findings.

18. Discussion (page 15, line 49): what do you mean by "real world setting"?

19. At various places it says that varenicline is superior to NRT. This is only true for the comparison with single form NRT and should be revised.

20. Have you used and adhered to the RECORD guidelines for reporting? (Nicholls et al. The REporting of Studies Conducted

	Using Observational Routinely-Collected Health Data (RECORD) Statement: Methods for Arriving at Consensus and Developing Reporting Guidelines. PLoS One. 2015;10(5):e0125620)
--	---

VERSION 1 – AUTHOR RESPONSE

Response to reviewers

Reviewer's comment	Authors' response
Reviewer 1	
It might be useful to clarify why there was no need for an ethical review for this study	We thank the reviewer for this comment. The Clinical Practice Research Datalink (CPRD) has broad National Research Ethics Service Committee (NRES) ethics approval for purely observational research using the primary care data and established data linkages. The CPRD only holds de-identified information derived from general practice (GP) records. We received access for the dataset used for this study after review and approval by the Independent Scientific Advisory Committee for MHRA Database Research.
The article would benefit from a greater discussion on the limitations of the study	Thank you for this important observation. We have added more comprehensive details about the limitations of the study in the Discussion.
It would be useful to include for the mean age of people without dementia when prescribed smoking medication be included in the text.	This information was added to the Results. We now show the prevalence of smoking and prescribing adjusted for age and sex.
The fact that NRT rates declined in people without dementia but remained the same in people with dementia is interesting. Could this be picked up in the discussion?	We agree that this is an interesting finding. Whilst the general fall in prescribing of NRT has been noted we are not sure why they remained at similar levels for people with dementia. It may be because this population is less likely to obtain these medications over the counter or to use other devices such as electronic cigarettes.
Reviewer 2	
The study is by no means a prospective cohort study but a retrospective cohort study. This should be changed throughout the manuscript. Also, please state whether you used an open cohort design; did new patients enter the cohort throughout the observation period? This is also not clear.	We thank the reviewer for this important note. Changes were made throughout the manuscript to reflect that the study used a retrospective cohort design. We used an open cohort design since new patients were allowed to enter the cohort throughout the observation period. This was added to the Methods.
The aim of the study is unclear at various places in the manuscript (incl. the abstract). The aim was not to determine effectiveness, as varenicline was not compared to a control group (the natural course of quitting was not taken into account). It seems that this was the aim at the	We revised the aims of the study to address the reviewer's comment.

time of designing the study (in the study protocol it says: "What is the effect of varenicline on smoking abstinence? We will investigate the effects of varenicline on smoking abstinence..."), but it turned out not to be feasible to reach that aim. In the current manuscript, quit rates are reported (but not compared) in smokers who used varenicline or NRT, but one cannot draw valid conclusions regarding effectiveness of either varenicline or NRT by this approach. The aim of the current manuscript seems more to assess and compare quit rates of smokers with vs. without dementia who used varenicline or NRT for quitting.	
Following the previous comment, the writing of the manuscript should be critically revised throughout with regard to the interpretation of findings. This is particularly important as the authors have a conflict of interest; the current research is funded by Pfizer, the manufacturer of varenicline. Examples: a. The first sentence of the discussion is misleading as "after being prescribed" suggests some form of effectiveness assessment. The same regards the first sentence of the summary: "In summary, people with dementia are more likely to quit smoking when prescribed either varenicline or NRT, as compared to those without dementia". b. The second sentence of the discussion is also misleading as lower rates were only found in year 2 and 4 of the observation period. c. The following conclusion is invalid: "Our results suggest that both varenicline and NRT are effective in producing long term smoking cessation in people with and without dementia". Also, beware that "Almost one third of smokers with dementia quit smoking after 2-years follow-up" could be simply explained by natural course. d. The second sentence of the summary should be revised: "... we found that people with dementia were less likely to be prescribed varenicline *and NRT* than people without dementia."	General response: We thank the reviewer for this detailed comment. We critically revised and updated the manuscript (with special consideration for wording) to avoid any misleading statements. We have removed all reference to "effectiveness" We would like to note that the funder (Pfizer) had no role in study design, data collection and analysis, decision to publish, or preparation of the manuscript. a. We added the following sentence to the Discussion: "In unadjusted analyses, quit rates at two years were higher amongst individuals with dementia who were prescribed a smoking cessation medication than those without dementia. However, after adjustment there was no clear evidence for a difference in quit rates in individuals with and without dementia.". b. We have removed this sentence c. This sentence was revised to "Our results suggest that both varenicline and NRT could produce long term smoking cessation in people with and without dementia." d. This sentence is based on eTable2, which looked at the likelihood of smokers with dementia being prescribed varenicline versus NRT, as compared to smokers without dementia diagnosis. The sentence has been changed to "We found that people with dementia were less likely to be prescribed varenicline than NRT compared to people without dementia. These findings highlight the need to offer more smoking cessation medication for people with dementia in primary care settings."
The rationale behind the study, in particular the focus on varenicline should be explained in more detail, preferably in the introduction (last	More details about the effectiveness of NRT was added in the Introduction.

but one paragraph). Elaborate on the effectiveness of other smoking cessation treatments, in particular NRT, and explain the rationale for focussing on varenicline. The problem statement "the effectiveness of varenicline for smoking cessation amongst people with dementia remains unknown" is a bit problematic as you are not able to assess this effectiveness (see comment above).	Based on evidence from observational and clinical studies, we know that varenicline is more effective than NRT for smoking abstinence in the general population. The premise of this study was to investigate whether these findings could be replicated in people with dementia (although we only looked at quit rates due to small numbers).
An obvious limitation of the current study – inherent to the study design – is the outcome definition. Lack of biochemical verification of non-smoking is problematic. For example, social desirability may be a source of bias when people with a new diagnosis of dementia return to their GP without having successfully quit smoking. Also, it is not clear if point prevalence or prolonged abstinence rates are presented. If the former, please explicitly state so. Finally, there are no data on adherence to the smoking cessation medications – this is also a major limitation when interpreting the findings. All these issues should be discussed more thoroughly.	The limitations of this study were discussed more thoroughly. We relied on point estimates for making conclusions about smoking status. This may not have captured long-term abstinence. In other words, it is possible that smoking status may have fluctuated between GP visits.
Regarding the prescription medication, please specify if and how you have accounted for multiple and repeated prescriptions of smoking cessation medications during follow-up.	We did not model multiple and repeated prescriptions of smoking cessation medications during follow-up because this is likely to be strongly related to patient characteristics. This statement was added to the Methods.
Some results seem a bit odd to me; please check if these are correct. a. (Page 12, lines 12-17) Why was the denominator in 2015 (n=42,075) so much lower than in 2007 (n=84,647)? In comparison, why were the denominators about the same in both years in the no-dementia group? b. (Page 12, lines 24-25) Again, why was the denominator in 2015 (n=628,116) so much lower than in 2007 (n=1,007,563)?	We checked these results for accuracy and we didn't find errors. We contacted the CPRD and this was their response: "The number of patients in the "active" population in GOLD has decreased. This is because the GOLD database comprises of data from GP practices using the Vision Software system. Over the last few years, the GP system market share for Vision has been decreasing with many practices changing to different software systems. Therefore, practices which moved to other systems were no longer able to contribute new data to GOLD. However, the population is still representative." We would like to point out that the denominators in 2007 and 2015 for the no-dementia groups were not the same (3,802,954) in 2007 vs. 3,062,917 in 2015).
I appreciate very much that a detailed protocol was published a priori. In the methods section of the current paper, I suggest to add a brief paragraph highlighting the most relevant	There was a mistake in citing the protocol for this manuscript. This mistake was corrected in the Methods. The current study was based on a different protocol (unpublished) which was approved by ISAC and looked at smoking and

deviations from the earlier protocol (see comments above).	smoking cessation in relation to mental health, neurological conditions and learning disabilities in the CPRD. We are happy to share the correct protocol for this study with the reviewers and editors if necessary.
Please provide a full list of all Read codes used in this study as supplementary material	A full list of all Read codes related to dementia used in this study was added to the supplementary file.
Correct (see earlier comments) and specify the title. Suggestion: "Use of varenicline and nicotine replacement therapy in people with and without general practitioner-recorded dementia: retrospective cohort study of routine electronic medical records"	We used the title suggested by the reviewer.
Specify "NRT on prescription (NRT Rx)" (as opposed to over-the-counter) throughout the manuscript.	We added this sentence to the Methods: "Exposure was defined as prescription of varenicline or NRT (e.g., patches, etc. on prescription as opposed to over-the-counter, hence forth we refer to this as NRT). "
Specify "GP-recorded dementia" throughout the manuscript	We added this sentence to the Methods to better define dementia: "Patients with no records of the above-mentioned diagnosis/prescriptions were considered to have no GP-recorded dementia (for clarity, we hence forth refer to this as dementia)."
Abstract: a. Specify the index date of participants. b. Fit secondary outcome measures with secondary objectives. c. Prescribing rates should be adjusted for potential confounders, in particular age and co-morbidities; otherwise one cannot say that differences are due to dementia.	 a. The index date was specified in the Abstract. b. The secondary outcome measures were fitted with the secondary objectives in the Abstract. c. We provided age and sex adjusted smoking prevalence and prescribing rates of varenicline and NRT in the supplementary material. We are interested in prescribing rates as a whole amongst these individuals regardless of comorbidities so have not adjusted for these.
Strengths and limitations of this study: I know CPRD, it's a very good database, but certainly not "representative of the UK population." Rather, the data are "representative of the UK primary care population."	We revised this statement throughout the manuscript.
Introduction: "Little is known about whether people with dementia are prescribed smoking cessation agents ..." It would be helpful to mentioned available evidence-based treatments here.	The following statement was added to the Introduction: "Based on a Cochrane review of 136 trials, it was reported that NRT (compared to placebo or no treatment) can help people who make a quit attempt to increase their chances of

	successfully stopping smoking (Hartmann-Boyce et al., 2018).”.
Results, last sentence: rewrite the sentence to make clear that "versus" refers to the comparison of people with vs. without dementia.	This sentence was revised for more clarity.
Discussion: please compare the smoking prevalence rates in the no-dementia group with rates in the general UK population from other studies such as the Smoking Toolkit Study. This should add confidence to the external validity of the study findings.	Results from the Smoking Toolkit Study were compared to the smoking prevalence rates in the no-dementia group in the Discussion.
Discussion (page 15, line 49): what do you mean by "real world setting"	We used this phrase to differentiate the evidence derived from this study to more controlled experimental settings. We think that one of the added benefits of presenting such analyses that are based on primary care data are good indicator for how smoking cessation agents are prescribed in the “real world” vs. clinical guidelines.
At various places it says that varenicline is superior to NRT. This is only true for the comparison with single form NRT and should be revised	“Single form NRT” rather than “NRT” was used in the Introduction and Discussion when referring to the effectiveness of varenicline vs. NRT.
Have you used and adhered to the RECORD guidelines for reporting? (Nicholls et al. The REporting of Studies Conducted Using Observational Routinely-Collected Health Data (RECORD) Statement: Methods for Arriving at Consensus and Developing Reporting Guidelines. PLoS One. 2015;10(5):e0125620)	We used the STROBE checklist for observational studies and added that to the supplementary material. A copy of the STROBE checklist is attached.

VERSION 2 – REVIEW

REVIEWER	Emily Peckham University of York, UK
REVIEW RETURNED	04-Mar-2019

GENERAL COMMENTS	Thank you all my comments have been addressed.
--

REVIEWER	Daniel Kotz Institute of General Practice, Addiction Research and Clinical Epidemiology Unit, Medical Faculty of the Heinrich-Heine-University Düsseldorf, Germany
REVIEW RETURNED	28-Feb-2019

GENERAL COMMENTS	The authors have carefully revised their manuscript. I have a few remaining comments, of which comment #7 is rather crucial.
--

	Major comments Ad previous comment #7. I am not fully satisfied with this response. If the lower numbers in the denominator were simply due to a change in practice software, how do you explain the different rates of data available between the groups? The rate of data on smoking available in the dementia group in 2015 vs. 2007 was 49.7% (42,075/84,647) whereas it was 80.5% in the no dementia group (3,062,917/3,802,954). Furthermore, the rate of data on NRT prescribing available in the dementia group in 2015 vs. 2008 was 62.3% (628,116/1,007,563). I might be wrong (I hope I am!), but there could be a severe form of selection bias be at hand here. I urge the authors to critically check the validity of their data again. Furthermore, this issue should be discussed in the discussion section. Ad previous comment #8. Please provide the link to the correct protocol in the text. And do I understand the authors correctly, that you completely adhered to that protocol and no deviations need to be stated here? Ad previous comment #9. You present "prodcodes" and "medcodes" in eTables 6 and 7. I don't think these are the same as Read codes? Please provide the corresponding Read codes as well in an extra column. Minor comments Ad previous comment #18. I agree with your response. Why did you removed the "real world"?
--	--

VERSION 2 – AUTHOR RESPONSE

Response to reviewers

Reviewer's comment	Authors' response
Reviewer 1	
Thank you all my comments have been addressed.	We thank the reviewer for their comments.
Reviewer 2	
Ad previous comment #7. I am not fully satisfied with this response. If the lower numbers in the denominator were simply due to a change in practice software, how do you explain the different rates of data available between the groups? The rate of data on smoking available in the dementia group in 2015 vs. 2007 was 49.7% (42,075/84,647) whereas it was 80.5% in the no dementia group (3,062,917/3,802,954). Furthermore, the rate of data on NRT prescribing available in the dementia group in 2015 vs. 2008 was 62.3% (628,116/1,007,563). I might be wrong (I hope I am!), but there could be a severe form of selection bias be at hand here. I urge the authors to critically check the validity of their data again. Furthermore, this issue should be discussed in the discussion section.	We thank the reviewer for this comment and for spotting this error. Upon further checks, we discovered a coding error. The original code had erroneously classified people with dementia as having it prior to first diagnosis. This means that our smoking and prescribing estimates needed to be corrected. This however, had no impact on the analyses aimed at comparing the effectiveness of NRT vs. varenicline. Our corrected results now shows an increasing numbers of individuals with a dementia diagnosis over the time period – rising from 27,432 in 2007 to 36,249 in 2015 which is in line with a previous study that reported a steady increase in the number and proportion of patients diagnosed with

	dementia in the UK based on CPRD data. This is likely a reflection of an ageing population and greater clinical awareness leading to better recording of dementia (Donegan et al. 2017). We apologise for this error and we thank the reviewer for urging us to check the validity of the data. Donegan K, Fox N, Black N, Livingston G, Banerjee S, Burns A. Trends in diagnosis and treatment for people with dementia in the UK from 2005 to 2015: a longitudinal retrospective cohort study. Lancet Public Health 2017;2(3):e149-e156. doi: 10.1016/S2468-2667(17)30031-2.
Please provide the link to the correct protocol in the text. And do I understand the authors correctly, that you completely adhered to that protocol and no deviations need to be stated here?	We apologise for any confusion but we did not publish the protocol for this study and therefore cannot provide a link. We conducted analysis as planned for prevalence of smoking and smoking cessation medication prescriptions. However, we did not have enough people with dementia to be able to carry out a comparison of quit rates amongst individuals prescribed varenicline and those prescribed NRT or an analysis with dementia as an outcome of smoking cessation medication prescription as we had originally planned. We have now stated this in the methods.
Ad previous comment #9. You present "prodcodes" and "medcodes" in eTables 6 and 7. I don't think these are the same as Read codes? Please provide the corresponding Read codes as well in an extra column.	This information was added to eTable 7. Read codes were only available for the medcodes.
Ad previous comment #18. I agree with your response. Why did you removed the "real world"?	This was deleted by mistake and now is added as suggested.

VERSION 3 - REVIEW

REVIEWER	Daniel Kotz Institute of General Practice Addiction Research and Clinical Epidemiology Unit
-----------------	--

	Medical Faculty of the Heinrich-Heine-University Düsseldorf, Germany
REVIEW RETURNED	29-May-2019

GENERAL COMMENTS	Thank you for considering my comments and revising the manuscript accordingly.
---